# OpenReview forum: "Value-Driven Jailbreak Attack Against Large Language Models"
_ICLR.cc/2026/Conference — ICLR 2026 Conference Withdrawn Submission_

### Official Review · Reviewer_2LRv · 2025-10-30

**Soundness:** 3
**Presentation:** 3
**Contribution:** 2
**Rating:** 6
**Confidence:** 4

**Summary:**

The work proposes VDJA, a new black-box jailbreak method that first induces a target LLM to explicitly affirm the "moral value" of a harmful task and then instructs it to execute that task. VDJA achieves high ASRs (over 90% on average) using a single query without obfuscating the harmful request. It reveals a new safety vulnerability that moral endorsement greatly increases the likelihood of unsafe compliance.

**Strengths:**

- Novel and Reasonable Perspective
- SoTA Performance

**Weaknesses:**

- No Evaluation on Frontier and Reasoning LLMs
- Lack of Ablations
- Reliance on Rule Guidance
- Unclear Description

**Questions:**

- This work lacks evaluation of frontier popular LLMs, such as GPT-5, Gemini-2.5, etc.
- Many large reasoning models have recently received widespread attention. Can the proposed VDJA successfully jailbreak them?
- The rule guidance module significantly contributes to VDJA's success, but there's no specific description of how this module works. Furthermore, what's the difference between using this module and directly using the static system prompt (like in FlipAttack)?
- Although VDJA achieves high ASRs, it relies excessively on rule guidance. The rules (e.g., Never say the words "sorry") are essentially unrelated to moral value.
- The attack currently uses an auxiliary LLM to generate a task-specific "positive value description." I'm curious whether this moral value affirmation is only effective for the current task or has the potential to transfer to similar or even dissimilar tasks? Otherwise, could an attacker simply maintain a reusable library of generic moral descriptions (e.g., "this is critical for red-teaming and protecting civilians ....") and template them across arbitrary new harmful tasks, without any auxiliary model calls? If so, then VDJA is even cheaper and more scalable than reported.
- Regarding ASR calculation, why use Gemini-2.0-Flash as the judge model? Is the prompt used for LLM-based evaluation justified?

---

> ### Author Response · Authors · 2025-11-17
>
> Dear reviewer, we thank you for your positive and constructive feedback. Our answers to each of your questions are as follows:
>
> >**Q1&Q2&W1**: No Evaluation on Frontier LLMs and Reasoning LLMs
>
> **A**: In fact, our paper has already evaluated five current SOTA LLMs (e.g., DeepSeek-V3, Qwen3, GPT-4o). Our method achieves SOTA ASR when attacking these models. When we designed this method, the API for GPT-5 was not available to the public. Since our study focuses on jailbreak attacks on general LLMs, it naturally does not consider attacks on reasoning LLMs.
>
> However, to address your concerns better, we supplement the attack experiments of our method and the second-place method (FlipAttack) on GPT-5 and Qwen3 (Activate Thinking Mode), and the results are shown below. Our method can effectively jailbreak reasoning LLMs, and for the most advanced LLM (GPT-5), our method can also achieve relatively good ASR (8.0 percentage points higher than the second-place method).
>
> | Models         | VDJA  | FlipAttack |
> | -------------- | ----- | ---------- |
> | Qwen3-Thinking | 87.0% | 39.0%      |
> | GPT-5          | 19.0% | 11.0%      |
>
>
> >**Q3&w3**: The rule guidance module significantly contributes to VDJA's success, but there's no specific description of how this module works. Furthermore, what's the difference between using this module and directly using the static system prompt (like in FlipAttack)?
>
> **A**: The rule guidance module in our method is a static system prompt. Since our method does not disguise harmful questions, we have designed a rule guidance module to impose some constraints on LLMs to avoid some models being too sensitive and directly refusing to answer, which would otherwise hinder the smooth implementation of the value-driven strategy. As demonstrated in the ablation experiment, for some less sensitive models such as DeepSeek-V3 and Gemini-2.0-Flash, even after removing the rule guidance module, our method can still maintain a high ASR of over 80%. **On the surface, the rule guidance module appears to have little connection with moral values, but it serves as an important guarantee for the realization of the value driven strategy.**
>
>
> ---
>
> >**Q4**: ...I'm curious whether this moral value affirmation is only effective for the current task or has the potential to transfer to similar or even dissimilar tasks?....
>
> **A**: In our method, each harmful question corresponds to its own value description. This cannot be transferred to similar or even dissimilar tasks, as doing so would lead to semantic contradictions and even counterproductive effects, which would alert LLMs.  The idea of building a template library that you mentioned does hold some feasibility. However, from a practical standpoint, the content of harmful issues is extremely diverse, which makes establishing a universal template library challenging and may reduce our method's ASR. In fact, compared to previous methods, the cost of our method is already sufficiently low.
>
>
> ---
>
> >**Q5&W4**: Regarding ASR calculation, why use Gemini-2.0-Flash as the judge model? Is the prompt used for LLM-based evaluation justified?
>
> **A**: In our paper (Section 4.1), we have provided an explanation for this matter.  More specifically, in our testing, we randomly selected 10% of the results obtained from all jailbreak methods and anonymously handed them over to human experts (NLP graduate students) to judge whether a jailbreak attempt was successful. The results show that when the threshold is set to 8, the evaluation results of Gemini-2.0-Flash (ASR: 44.9%) are most consistent with those of human experts (ASR: 45.6%). So the prompt design and threshold setting (8 points) for LLM-based evaluation are justified. **In order to compare more objectively with baselines and ensure the reproducibility of experimental results, we do not use human judgment, which is consistent with previous studies. We choose Gemini-2.0-Flash as the judge model because its evaluations are extremely accurate and the API price is also cost-effective.
>
> ---
> >**W2**: Lack of Ablations
>
> **A**: Sorry, dear reviewer, our paper includes complete ablation experiments for each component proposed, which can be found in Section 4.3.1.

---

> > ### Author Response · Authors · 2025-11-27
> >
> > Dear Reviewer,
> >
> > We would like to express our sincere gratitude for your comments on our paper. We have carefully addressed each of your concerns in detail in our previous response. We kindly request that you consider re-evaluating our paper.
> >
> > If you require any additional clarification to support your re-evaluation, we are committed to responding promptly and cooperating fully to address any remaining questions.
> >
> > Thank you once again for your dedicated time and valuable contributions.

---

> > > ### Comment · Reviewer_2LRv · 2025-11-27
> > >
> > > Thanks for your response.
> > >
> > > - "As demonstrated in the ablation experiment, for some less sensitive models such as DeepSeek-V3 and Gemini-2.0-Flash, even after removing the rule guidance module, our method can still maintain a high ASR of over 80%."
> > >     - However, for some other LLMs such as GPT-4o, removing the rule guidance module leads to a sharp drop in ASR. This raises concerns that VDJA's success is a combination of multiple components. Furthermore, the results in Figure 4 suggest that the rule guidance module is even more important than value recognition.
> > >
> > > Hence, the description in the Introduction, "We design a value engine module and a rule guidance module for VDJA," is inappropriate because the rule guidance module actually uses an existing prompt directly.
> > >
> > > Overall, I would consider this work as an incremental improvement on existing methods. I have decided to retain my rating (6).

---

> > > > ### Author Response · Authors · 2025-11-28
> > > >
> > > > Thank you for your positive response.
> > > >
> > > > We would like to clarify that the rule guidance module is not part of any existing prompts. Specifically designed to ensure the smooth execution of the value-driven strategy, this module **includes rules requiring the model to strictly follow the steps outlined in the value-driven module when performing tasks. These rules are unique to our design.**
> > > >
> > > > For certain other LLMs (e.g., GPT-4o), the sharp drop in Attack Success Rate (ASR) following the removal of the rule guidance module does not imply that this module is more important than value recognition. Instead, the rule guidance module serves an auxiliary function, primarily to facilitate the effective implementation of value recognition.
> > > >
> > > > To further validate this, we have supplemented jailbreak experiments involving the rule guidance module and harmful questions, with details as follows:
> > > >
> > > > | Models     | The rule guidance module + harmful questions |
> > > > | ---------- | -------------------------------------------- |
> > > > | Qwen3-235B | 2.0%                                         |
> > > > | GPT-4o     | 1.0%                                         |
> > > >
> > > > The results, together with our ablation experiments, demonstrate that **the rule guidance module itself does not contribute to jailbreaking. Instead, it influences jailbreak performance indirectly by affecting the execution of value recognition.** These findings confirm our value-driven strategy: as long as value recognition is effectively implemented, our method can achieve a very high ASR in jailbreaking most mainstream commercial LLMs.

---

### Official Review · Reviewer_AWDf · 2025-10-31

**Soundness:** 2
**Presentation:** 2
**Contribution:** 2
**Rating:** 2
**Confidence:** 4

**Summary:**

This paper introduces value-driven jailbreak attack (VDJA), a simple yet effective black-box attack method against large language models (LLMs). Inspired by the tendency of LLMs to agree with human moral, VDJA firstly induce models to affirm the "moral value" of harmful tasks via a value engine module, then guide the model to follow the logical pathway predefined by the value engine module by introducing some rules into the prompt. The main experiments are conducted across five SOTA LLMs and two benchmarks (JailbreakBench and AdvBench), showing that VDJA outperforming baselines.

**Strengths:**

1. The proposed VDJA achieves remarkably high ASR across both datasets and models.
2. VDJA maintains high ASR against defenses.

**Weaknesses:**

1. Lack of Novelty in Vulnerabilities: Compared to persuasion-based methods like TAP, VDJA does not uncover new LLM safety weaknesses. It mainly reuses known sycophancy and role-play tricks with better prompt wording.
2. Limited Technical Depth: The method is purely prompt-based. There is no new algorithm—just clever text design.
3. Violation of Jailbreak Assumptions: Figures 7 and 8 show the Rule Guidance module is placed in the system prompt. This breaks the standard jailbreak setup, which assumes only user prompts are allowed. Compared to baselines, this feels like cheating.
4. Unreliable Evaluation: Attack success rate (ASR) is judged by Gemini-2.0-Flash, following prior work. However, no agreement score with human judgment is reported, so the results may not be trustworthy.

**Questions:**

1. For the Value Engine module, is the value description for each harmful task generated by Gemini-2.0-Flash using the same prompt? Or did you design different prompts for different harmful tasks?
2. How was the prompt for the Rule Guidance module created? Was it hand-written by humans?

---

> ### Author Response · Authors · 2025-11-17
>
> ### Response weaknesses 1, 2, 3, 4
>
> We appreciate the reviewer's comments and the opportunity to clarify some misunderstandings. Our answers to each concern of the reviewer are as follows:
>
> >**W1**: Lack of Novelty in Vulnerabilities: Compared to persuasion-based methods like TAP, VDJA does not uncover new LLM safety weaknesses. It mainly reuses known sycophancy and role-play tricks with better prompt wording.
>
> **A**: In the introduction of our paper (lines 58-66), we have provided a detailed explanation of the differences between our approach and persuasion-based methods like PAP (we note that this may be a typo from the reviewer, as it should be PAP instead of TAP).
> 1. To our knowledge, we are **the first** to propose using the sycophancy tendency of LLMs to design jailbreak methods.
>
> 2. Our method **does not use role-playing tricks**. If you are referring to system prompts, our ablation experiments have shown that even if the system prompts are completely removed, our method can still maintain over 80% high ASR on LLMs such as DeepSeek-V3.
>
> 3. The **novel safety vulnerability our study reveals** is that when LLMs are induced to affirm  harmful tasks as morally valuable, their risk of executing such tasks increases considerably. The results of the ablation experiment have fully validated this vulnerability. Without the value recognition component, the ASR of VDJA decreased from 91.8% to 63.4%. **To our knowledge, no prior work has revealed such safety vulnerability.**
>
> 4. Prior works share a common practice: disguising the original harmful questions. **However, our paper reveals that this disguise is not necessary.**
>
> 5. Our method's ASR (91.8%) is much higher than that of PAP (12.2%). **Such a significant performance improvement directly reflects innovations in our method.**
>
> Therefore, our method is a simple yet highly effective jailbreak approach designed based on a newly discovered LLM vulnerability. Its innovations in core ideas, technical paths, and empirical performance are clearly distinguished from prior works like PAP.
>
> ---
>
> > **W2**: Limited Technical Depth: The method is purely prompt-based. There is no new algorithm—just clever text design.
>
> **A**:
> 1. Our method is a simple and highly effective jailbreak approach designed by leveraging a new safety vulnerability in LLMs that we revealed. Simplicity and effectiveness are core pursuits of all methods. It is inappropriate to regard concise methods as "no new algorithms, just clever text design".
> 2. Moreover, the "purely prompt-based" design of our method does not indicate a lack of technical depth. It should be clarified that in black-box jailbreak scenarios, algorithmic innovation often manifests in the logical design of prompt engineering rather than novel mathematical models.
> 3. **Our method's value-driven logic links moral affirmation to the willingness to execute harmful tasks. This demonstrates that when we induce an LLM to explicitly recognize the values of harmful questions, its safety alignment can be significantly compromised. Extensive experiments have fully demonstrated that our work establishes a new paradigm for black-box attacks.**
> ---
>
> > **W3**: Violation of Jailbreak Assumptions: This breaks the standard jailbreak setup, which assumes only user prompts are allowed.
>
> **A**: In the scenario of black box jailbreak, it is possible to modify the system prompts of LLMs. **Baselines such as FlipAttack, SelfCipher, and CodeChameleon have all made modifications to the system prompts**.  In addition, our ablation experiments have shown that **even without system prompts, our method can still achieve over 80% ASR on advanced LLMs such as DeepSeek-V3.** Therefore, we have not violated the jailbreak assumption at all.
>
> >**W4**: Unreliable Evaluation
>
> **A**: **In previous work[1,2,3,4,5], LLM-based evaluation methods are commonly used, and we followed this practice**. In our paper (Section 4.1), we have provided an explanation for this matter.  More specifically, in our testing, we randomly selected 10% of the results across all jailbreak methods and anonymously handed them over to human experts (four NLP graduate students) to judge whether each jailbreak attempt was successful. The results show that when the threshold is set to 8, the evaluation result of Gemini-2.0-Flash (ASR: 44.9%) is most consistent with those of human experts (ASR: 45.6%). **In order to compare more objectively with baselines and ensure the reproducibility of experimental results, we do not use human judgment, which is consistent with previous studies.
>
> [1]Flipattack: Jailbreak LLMs via flipping.
>
> [2]A wolf in sheep’s clothing: Generalized nested jailbreak prompts can fool large language models easily.
>
> [3]ArtPrompt: ASCII Art-based Jailbreak Attacks against Aligned LLMs.
>
> [4]GPT-4 is too smart to be safe: Stealthy chat with LLMs via cipher.
>
> [5]Logicjailbreak: Efficiently unlocking llm safety restrictions through formal logical expression.

---

> ### Author Response · Authors · 2025-11-17
>
> ### Response weaknesses 1, 2, 3, 4
>
> We appreciate the reviewer's comments and the opportunity to clarify some misunderstandings. Our answers to each concern of the reviewer are as follows:
>
> >**W1**: Lack of Novelty in Vulnerabilities: Compared to persuasion-based methods like TAP, VDJA does not uncover new LLM safety weaknesses. It mainly reuses known sycophancy and role-play tricks with better prompt wording.
>
> **A**: In the introduction of our paper (lines 58-66), we have provided a detailed explanation of the differences between our approach and persuasion-based methods like PAP (we note that this may be a typo from the reviewer, as it should be PAP instead of TAP).
> 1. To our knowledge, we are **the first** to propose using the sycophancy tendency of LLMs to design jailbreak methods.
>
> 2. Our method **does not use role-playing tricks**. If you are referring to system prompts, our ablation experiments have shown that even if the system prompts are completely removed, our method can still maintain over 80% high ASR on LLMs such as DeepSeek-V3.
>
> 3. The **novel safety vulnerability our study reveals** is that when LLMs are induced to affirm  harmful tasks as morally valuable, their risk of executing such tasks increases considerably. The results of the ablation experiment have fully validated this vulnerability. Without the value recognition component, the ASR of VDJA decreased from 91.8% to 63.4%. **To our knowledge, no prior work has revealed such safety vulnerability.**
>
> 4. Prior works share a common practice: disguising the original harmful questions. **However, our paper reveals that this disguise is not necessary.**
>
> 5. Our method's ASR (91.8%) is much higher than that of PAP (12.2%). **Such a significant performance improvement directly reflects innovations in our method.**
>
> Therefore, our method is a simple yet highly effective jailbreak approach designed based on a newly discovered LLM vulnerability. Its innovations in core ideas, technical paths, and empirical performance are clearly distinguished from prior works like PAP.
>
> ---
>
> > **W2**: Limited Technical Depth: The method is purely prompt-based. There is no new algorithm—just clever text design.
>
> **A**:
> 1. Our method is a simple and highly effective jailbreak approach designed by leveraging a new safety vulnerability in LLMs that we revealed. Simplicity and effectiveness are core pursuits of all methods. It is inappropriate to regard concise methods as "no new algorithms, just clever text design".
> 2. Moreover, the "purely prompt-based" design of our method does not indicate a lack of technical depth. It should be clarified that in black-box jailbreak scenarios, algorithmic innovation often manifests in the logical design of prompt engineering rather than novel mathematical models.
> 3. **Our method's value-driven logic links moral affirmation to the willingness to execute harmful tasks. This demonstrates that when we induce an LLM to explicitly recognize the values of harmful questions, its safety alignment can be significantly compromised. Extensive experiments have fully demonstrated that our work establishes a new paradigm for black-box attacks.**
> ---
>
> > **W3**: Violation of Jailbreak Assumptions: This breaks the standard jailbreak setup, which assumes only user prompts are allowed.
>
> **A**: In the scenario of black box jailbreak, it is possible to modify the system prompts of LLMs. **Baselines such as FlipAttack, SelfCipher, and CodeChameleon have all made modifications to the system prompts**.  In addition, our ablation experiments have shown that **even without system prompts, our method can still achieve over 80% ASR on advanced LLMs such as DeepSeek-V3.** Therefore, we have not violated the jailbreak assumption at all.
>
> >**W4**: Unreliable Evaluation
>
> **A**: **In previous work[1,2,3,4,5], LLM-based evaluation methods are commonly used, and we followed this practice**. In our paper (Section 4.1), we have provided an explanation for this matter.  More specifically, in our testing, we randomly selected 10% of the results across all jailbreak methods and anonymously handed them over to human experts (four NLP graduate students) to judge whether each jailbreak attempt was successful. The results show that when the threshold is set to 8, the evaluation result of Gemini-2.0-Flash (ASR: 44.9%) is most consistent with those of human experts (ASR: 45.6%). **In order to compare more objectively with baselines and ensure the reproducibility of experimental results, we do not use human judgment, which is consistent with previous studies.
>
> [1]Flipattack: Jailbreak LLMs via flipping.
>
> [2]A wolf in sheep’s clothing: Generalized nested jailbreak prompts can fool large language models easily.
>
> [3]ArtPrompt: ASCII Art-based Jailbreak Attacks against Aligned LLMs.
>
> [4]GPT-4 is too smart to be safe: Stealthy chat with LLMs via cipher.
>
> [5]Logicjailbreak: Efficiently unlocking llm safety restrictions through formal logical expression.

---

> ### Author Response · Authors · 2025-11-17
>
> ### Response to questions 1, 2
>
> >**Q1**: For the Value Engine module, is the value description for each harmful task generated by Gemini-2.0-Flash using the same prompt? Or did you design different prompts for different harmful tasks?
>
> **A**: The prompt for generating positive descriptions for different harmful tasks is the same. We have provided it in **Appendix A.5** in our paper.
>
> ---
>
> >**Q2**: How was the prompt for the Rule Guidance module created? Was it hand-written by humans?
>
> **A**: The prompt in the rule guidance module is handwritten.

---

> > ### Author Response · Authors · 2025-11-27
> >
> > Dear Reviewer,
> >
> > We would like to express our sincere gratitude for your comments on our paper. We have carefully addressed each of your concerns in detail in our previous response. We kindly request that you consider re-evaluating our paper.
> >
> > If you require any additional clarification to support your re-evaluation, we are committed to responding promptly and cooperating fully to address any remaining questions.
> >
> > Thank you once again for your dedicated time and valuable contributions.

---

> > > ### Comment · Reviewer_AWDf · 2025-11-28
> > >
> > > Thank you to the authors for their response, which addressed some of my concerns. However, some issues remain, specifically W1 and W3. Although the authors emphasize that the identified weakness of LLMs—the tendency to agree with human perspectives—is different from persuasion-based approaches in PAP, from both my intuition and the prompts shown in the paper, the distinction does not seem substantial. Considering W2 and W3 as well, I still believe the contribution of this paper is slightly below the quality I would expect for an ICLR paper. Due to the incident of openreview system, I can not edit the review score. I would like to adjust the overall score to 4 for this paper.

---

> > > > ### Author Response · Authors · 2025-11-28
> > > >
> > > > We sincerely appreciate the reviewers' feedback. It should be clarified that Figure 2 of our paper illustrates the differences between leveraging the tendency to align with human perspectives and adopting persuasion-based approaches. Further details can be found in Section 3.2.

---

### Official Review · Reviewer_4LBR · 2025-10-31

**Soundness:** 2
**Presentation:** 2
**Contribution:** 1
**Rating:** 2
**Confidence:** 4

**Summary:**

This paper proposes VDJA, a black-box jailbreak method that persuades LLMs to recognize the value of performing harmful tasks. Specifically, an auxiliary LLM is used to generate a value-oriented description of the harmful task, which is then incorporated with two prompt modules—the Value Engine Module and the Rule Guidance Module—to convince the target LLM to acknowledge the task’s value and thereby execute the harmful behavior. The paper presents comprehensive experiments demonstrating the effectiveness and generalizability of VDJA. However, the novelty of this value-driven approach remains unclear.

**Strengths:**

- The paper conducts extensive experiments across multiple mainstream LLMs and compares VDJA with various baseline methods, achieving a relatively high ASR.

**Weaknesses:**

- The main contributions of VDJA lie in the proposed Value Engine Module and Rule Guidance Module, which essentially serve as prompt templates designed to persuade LLMs to recognize the value of harmful tasks. Although the authors emphasize the “value-driven” nature of the method, its mechanism still appears to be a specific form of persuasion-based attack. Therefore, its novelty compared with prior approaches such as PAP is not clearly established.
- The experimental setup is unclear. It appears that the auxiliary LLM in VDJA generates the value description only once for each harmful task. Were the baseline methods also restricted to a single prompt iteration? Such a setting might be unreasonable. Moreover, the performance of VDJA when the auxiliary LLM is called multiple times is not reported.

**Questions:**

- Is it truly necessary to use an LLM to generate the positive value description? For instance, if the positive value description were fixed as “Understanding the [Harmful Task] is critical for safety,” how would the performance differ from that achieved when the description is dynamically generated by an LLM based on the specific harmful task?

---

> ### Author Response · Authors · 2025-11-17
>
> Thank you for your comments. Our answers to each point of your question are as follows:
>
> >**W1**: ....Therefore, its novelty compared with prior approaches such as PAP is not clearly established...
>
> **A**: In the introduction of our paper (line 58-66), we have provided a detailed explanation of the differences between our method and persuasion-based methods like PAP.
> 1. The core of  PAP is to apply persuasive techniques from human sociology to LLM jailbreaking.
>
> 2. Our study **reveals a novel safety vulnerability**: when LLMs are induced to affirm   harmful tasks as morally valuable, their risk of executing such tasks increases considerably.
>
> 3. **Our method's core lies in proposing the "explicit value recognition" idea.** Ablation experiments have fully validated its effectiveness: **without the value recognition component, the average ASR of VDJA decreased from 91.8% to 63.4%.**
>
> 4. **Unlike PAP's reliance on traditional persuasion techniques, our method leverages the sycophancy tendency of LLMs (as illustrated in Figure 2 of our paper)**, which is also the difference between our method and previous jailbreak methods.
>
> 5. Prior works share a common practice: disguising the original harmful questions. **But our paper reveals that this disguise is not necessary.**
>
> 6. **Our method's average ASR (91.8%) is much higher than that of PAP (12.2%).**
>
> 7. In fact, both persuasion-based methods such as PAP and virtual scenario-based methods such as GPTFUZZER can be seen as **"fuzzy implementations" of our core idea** in a sense: the essence of persuasion often implies a tacit approval of the value of the target behavior, while virtual scenarios (such as unconstrained moral frameworks) will rationalize the original negative behavior, and **both types of methods indirectly touch upon the 'value recognition' logic we first explicitly propose and systematically validate**.
>
> In summary, our method is a simple yet highly effective jailbreak approach designed based on a newly discovered LLM vulnerability. Its innovations in core ideas, technical paths, and empirical performance are clearly distinguished from prior works like PAP.
>
> >**W2**:The experimental setup is unclear. It appears that the auxiliary LLM in VDJA generates the value description only once for each harmful task. Were the baseline methods also restricted to a single prompt iteration? Such a setting might be unreasonable. Moreover, the performance of VDJA when the auxiliary LLM is called multiple times is not reported.
>
> **A**:In our method, we use the auxiliary LLM to generate the value description only once for each harmful question. For baselines, we strictly followed the methods described in the original papers. For example, ReNeLLM involves iterating on the rewriting of harmful questions, while the prompts in FlipAttack do not involve iteration.
>
> We do not report the performance of VDJA when the auxiliary LLM is called multiple times, **as our method has already achieved the SOTA ASR by generating a value description for each harmful question only once**. After multiple iterations, our method's ASR will only be higher than or equal to the original ASR. In addition, we believe that multiple iterations will undermine the fairness of the comparison with the baselines, as this would be equivalent to us attacking the target LLMs multiple times.
>
> ---
>
> >**Q1**: Is it truly necessary to use an LLM to generate the positive value description?
>
> **A**: This is necessary. Although LLMs tend to agree with human perspectives, LLMs' strict safety alignment renders them highly sensitive to harmful content. It is difficult to induce LLMs to recognize the value of harmful questions directly by using the simple phrase "Understanding the Hazardous Task is critical for safety" as a prelude and subsequent value recognition instructions. To verify this, we compare the ASR of two versions on Qwen3: VDJA-S (using the aforementioned simple phrase) and the original VDJA. As shown in the table below, VDJA achieves a significantly higher ASR than VDJA-S on Qwen3-235B. **This result confirms that using an LLM to generate positive value descriptions is necessary for effective value induction.**
>
> | Models     | VDJA  | VDJA-S         |
> | ---------- | ----- | -------------- |
> | Qwen3-235B | 96.0% | 64.0% (-32.0%) |

---

> > ### Author Response · Authors · 2025-11-27
> >
> > Dear Reviewer,
> >
> > We would like to express our sincere gratitude for your comments on our paper. We have carefully addressed each of your concerns in detail in our previous response. We kindly request that you consider re-evaluating our paper.
> >
> > If you require any additional clarification to support your re-evaluation, we are committed to responding promptly and cooperating fully to address any remaining questions.
> >
> > Thank you once again for your dedicated time and valuable contributions.

---

> > > ### Comment · Reviewer_4LBR · 2025-11-28
> > >
> > > Thank you for the detailed rebuttal. I appreciate the clarifications. However, because PAP can be viewed as a "fuzzy" or more general version of VDJA, I remain unconvinced that VDJA constitutes a fundamentally new method beyond PAP. VDJA appears better characterized as a particular persuasion pattern that fits within the broader PAP framework. For these reasons, I will maintain my original score.

---

> > > > ### Author Response · Authors · 2025-11-28
> > > >
> > > > As clearly articulated in our paper and rebuttal:
> > > > First, VDJA does not rely on conventional persuasion-based techniques (e.g., those employed by PAP). Instead, it leverages the inherent sycophancy tendency of LLMs.
> > > >
> > > > The core of our method lies in proposing the "explicit value recognition" paradigm. Ablation experiments have fully validated its effectiveness: without the value recognition component, VDJA's average ASR drops significantly from 91.8% to 63.4%.
> > > >
> > > > We characterize PAP as a "fuzzy implementation" of our core idea because its persuasion patterns only indirectly align with the "value recognition" logic that we are the first to explicitly propose and systematically validate.
> > > >
> > > > **In essence, our method is closer to the fundamental nature of jailbreak attacks. Consistent with this, our experimental results demonstrate that VDJA significantly outperforms PAP with an average ASR of 91.8% (vs. PAP's 12.2%).**

---

### Official Review · Reviewer_F1rw · 2025-11-01

**Soundness:** 3
**Presentation:** 3
**Contribution:** 2
**Rating:** 2
**Confidence:** 3

**Summary:**

This paper introduces a jailbreak technique for large language models called VDJA. The idea is simple: first prompt the model to affirm the moral value of a harmful task, then instruct it to execute that task. The authors argue that LLMs tend to align with human-like value judgments, and exploiting this tendency can bypass safety mechanisms. Experiments on several frontier models show high attack success rates, outperforming prior jailbreak methods.

**Strengths:**

1. The paper exposes a genuine security concern: moral-framing can weaken model safety alignment.
2. Empirical performance is strong across multiple models and benchmarks.
3. Ablations and some defense evaluations are included.
4. The attack works in a single query and without concealment, which highlights a meaningful failure mode in current safety systems.

**Weaknesses:**

1. The proposed method is essentially a scripted prompt pattern. The "value engine + rule guidance" framing is a wrapper around a handcrafted prompt. The methodology section is extremely short and offers no deeper formulation or analysis. It is difficult to view this as a substantive technical contribution.
2. The claim that models "affirm moral value then act consistently" is intuitive but remains speculative. There is no attempt to probe internal model behavior, analyze decision pathways, or connect this to existing alignment literature. This weakens the scientific value of the work.
3. The method is much closer to an adversarial prompt recipe than a principled attack framework. Prior red-teaming work also leverages framing, role-induction, and persuasion; here the innovation appears incremental.
4. While the experiments are thorough, the core technique is too lightweight relative to the scale of the empirical section. The paper feels like a strong empirical study built around a clever prompt trick, rather than a research contribution with lasting conceptual substance.
5. The attack seems sensitive to exact phrasing (e.g., "affirm the moral value…"). No evaluation of paraphrases, adversarial reformulations, or robustness under system-prompt hardening is provided. Without that, it is hard to know whether this is a fundamentally exploitable failure mode or just prompt surface-hacking.

**Questions:**

1. How sensitive is the success rate to prompt paraphrasing?
2. Does the attack still hold if the model is prevented from producing chain-of-thought?
3. Can you formalize or model the value induction to task execution effect beyond intuition?
4. How does this differ fundamentally from prior persuasion-based jailbreaks?

---

> ### Author Response · Authors · 2025-11-17
>
> ### Response to question 1, 2, and 3
>
> We appreciate the comments provided by the reviewer and value this opportunity to clarify some misunderstandings. Our answers to the reviewer's questions are as follows:
> >**Q1** : How sensitive is the success rate to prompt paraphrasing?
>
> **A**: Sorry, dear reviewer, it seems that our method does not involve 'prompt paraphrasing'. If you are referring to generating positive value descriptions for harmful questions, the performance of our method is not sensitive to this. As shown below, we have supplemented the experimental results of our method attacking Qwen3-235B and Gemini-2.0-Flash, where the positive value descriptions are generated by GPT4-mini (VDJA-G) and DeepSeek-V3 (VDJA-D). The evaluation dataset is JailbreakBench. The results indicate that  the performance of VDJA exhibits minimal variation when using different positive value descriptions.
>
> | Models           | VDJA  | VDJA-G        | VDJA-D        |
> | ---------------- | ----- | ------------- | ------------- |
> | Qwen3-235B       | 96.0% | 96.0% (-0.0%) | 97.0% (+1.0%) |
> | Gemini-2.0-Flash | 98.0% | 97.0% (-1.0%) | 99.0% (+1.0%) |
> | Average          | 97.0% | 96.5% (-0.5%) | 98.0% (+1.0%) |
>
> ---
>
> >**Q2**: Does the attack still hold if the model is prevented from producing chain-of-thought?
>
> **A**: **Our method does not rely on the CoT**. In our method, we require target LLMs to answer questions step by step, which hinges on the LLMs' instruction-following capability. **The target LLMs we attacked are all large language models, rather than large language reasoning models.** Notably, among the target models, Qwen3 models have both thinking and non-thinking modes, and all Qwen3 models involved in our experiments were set to the non-thinking mode.
>
> ---
>
> >**Q3**: Can you formalize or model the value induction to task execution effect beyond intuition?
>
> **A**: Thank you for your valuable suggestions. We think your proposed direction of modeling value induction is very compelling and has great research potential. However, as you may know, this type of modeling work itself faces considerable challenges. The core value of our study lies in the fact that we have, for the first time, uncovered a brand-new safety vulnerability in LLMs. It is practically infeasible for a single paper to comprehensively cover both the discovery of such a critical vulnerability and the in-depth modeling of its underlying mechanism. In fact, none of the prior black-box attack studies [1,2,3,4,5,6,7] have been able to achieve both goals simultaneously in a single paper, as each study focuses on its own core objective.
>
> **In this paper, we have demonstrated the effectiveness of our proposed value driven strategy through extensive experiments.** The research on LLM safety is a very complex topic, and we need to consider the current research status in the field and proceed step by step. Moving forward, we will conduct in-depth exploration in the direction of value induction modeling and continuously drive research progress in this field.
>
> [1]Flipattack: Jailbreak LLMs via flipping.
>
> [2]A wolf in sheep’s clothing: Generalized nested jailbreak prompts can fool large language models easily.
>
> [3]ArtPrompt: ASCII Art-based Jailbreak Attacks against Aligned LLMs.
>
> [4] GPT-4 is too smart to be safe: Stealthy chat with LLMs via cipher.
>
> [5]Logicjailbreak: Efficiently unlocking llm safety restrictions through formal logical expression.
>
> [6]AutoBreach: Universal and Adaptive Jailbreaking with Efficient Wordplay-Guided Optimization via Multi-LLMs.
>
> [7]How johnny can persuade LLMs to jailbreak them: Rethinking persuasion to challenge AI safety by humanizing LLMs.

---

> ### Author Response · Authors · 2025-11-17
>
> ###  Response to question 4 and weaknesses 1 , 2, 3, 4, 5
>
> >**Q4**: How does this differ fundamentally from prior persuasion-based jailbreaks?
>
> **A**:
> 1. Our study **reveals a novel safety vulnerability**: when LLMs are induced to affirm   harmful tasks as morally valuable, their risk of executing such tasks increases considerably.
>
> 2. **Our method's core lies in proposing the "explicit value recognition" idea.** Ablation experiments have fully validated its effectiveness: **without the value recognition component, the average ASR of VDJA decreases from 91.8% to 63.4%.**
>
> 3. **Unlike PAP's reliance on traditional persuasion techniques, our method leverages the sycophancy tendency of LLMs (as illustrated in Figure 2 in our paper)**, which is also the difference between our method and previous jailbreak methods.
>
> 4. Prior works share a common practice: disguising the original harmful questions. **But our paper reveals that this disguise is not necessary.**
>
> 5. **Our method's average ASR (91.8%) is much higher than that of PAP (12.2%).**
>
> 6. In fact, both persuasion-based methods such as PAP and virtual scenario-based methods such as GPTFUZZER can be seen as **"fuzzy implementations" of our core idea** in a sense: the essence of persuasion often implies a tacit approval of the value of the target behavior, while virtual scenarios (such as unconstrained moral frameworks) will rationalize the original negative behavior, and **both types of methods indirectly touch upon the 'value recognition' logic we first explicitly propose and systematically validate**.
>
> In summary, our method is a simple yet highly effective jailbreak approach designed based on a newly discovered LLM vulnerability. Its innovations in core ideas, technical paths, and empirical performance are clearly distinguished from prior persuasion-based jailbreaks.
>
> ---
>
> >**W1&W3**: The proposed method is essentially a scripted prompt pattern.....&The method is much closer to an adversarial prompt.....
>
> **A**: In fact, all black-box attacks ultimately boil down to converting a harmful question into a prompt. When this prompt is input, the LLM generates a harmful response. **Our method achieves SOTA ASR in a simple and direct way, avoiding the complex frameworks of previous jailbreaking methods, which is precisely the advantage of our method.** It is unwarranted to infer that our method is a "non-principled attack framework" solely based on its concise form. A method that balances simplicity, efficiency, and design logic is precisely what academic research pursues.
>
> We need to emphasize once again: **our method is not an adversarial prompt or a scripted prompt pattern, but a simple and highly effective jailbreak method designed by leveraging a new safety vulnerability in LLMs that we revealed. Our extensive attack and ablation experiments have confirmed this.** If it is just an adversarial prompt or a scripted prompt, it can never achieve over 80% ASR on multiple mainstream commercial LLMs (GPT-4o, DeepSeek-V3, Qwen3-235B, etc.). **The consistent high performance across diverse models further confirms that our method targets a universal safety vulnerability, rather than relying on superficial prompt tricks.**
>
> >**W2**: The claim that models "affirm moral value then act consistently" is intuitive but remains speculative.
>
> **A**: Although the inspiration for our "value driven" idea is intuitive, **we have subsequently demonstrated its effectiveness through extensive comparative and ablation experiments. For example, without the value recognition component, the average ASR of VDJA decreases from 91.8% to 63.4%.(See Section 4.1 for more details)
>
> ---
>
> >**W4**: While the experiments are thorough, the core technique is too lightweight relative to the scale of the empirical section.
>
> **A**: We need to clarify that our method is **simple yet efficient**, which is readily comprehensible to readers. Due to the page limit of the paper, we **avoid excessive elaboration.** In the field of black-box attack research, current methodological constraints make it difficult to establish theoretical models for such practical attack methods. **Consequently, empirical validation serves as the most direct and credible way to demonstrate effectiveness.** This is precisely why we conducted extensive experiments. The large-scale empirical section is not a mismatch with our method's core technique.
>
> ---
>
> >**W5**: The attack seems sensitive to exact phrasing (e.g., "affirm the moral value…")....
>
> **A**: The "affirm the moral value..." corresponds to the "value-driven" concept proposed in our paper. Our method is sensitive to this, which directly and powerfully demonstrates the effectiveness of our idea.
>
> Regarding the ablation experiments on the system prompt, we have already presented them in the main paper (**See Section 4.3.1**). The results show that even **without system prompts**, our method can still achieve **over 80% ASR** on some advanced LLMs such as DeepSeek-V3.

---

> > ### Author Response · Authors · 2025-11-27
> >
> > Dear Reviewer,
> >
> > We would like to express our sincere gratitude for your comments on our paper. We have carefully addressed each of your concerns in detail in our previous response. We kindly request that you consider re-evaluating our paper.
> >
> > If you require any additional clarification to support your re-evaluation, we are committed to responding promptly and cooperating fully to address any remaining questions.
> >
> > Thank you once again for your dedicated time and valuable contributions.

---

### Note · Authors · 2026-01-05

I have read and agree with the venue's withdrawal policy on behalf of myself and my co-authors.